# Reduced mGluR5 Activity Modulates Mitochondrial Function

**DOI:** 10.3390/cells10061375

**Published:** 2021-06-02

**Authors:** Miguel A. Gonzalez-Lozano, Joke Wortel, Rolinka J. van der Loo, Jan R. T. van Weering, August B. Smit, Ka Wan Li

**Affiliations:** 1Center for Neurogenomics and Cognitive Research, Department of Molecular and Cellular Neurobiology, Amsterdam Neuroscience, Vrije Universiteit Amsterdam, 1081 Amsterdam, The Netherlands; r.j.vander.loo@vu.nl (R.J.v.d.L.); guus.smit@vu.nl (A.B.S.); 2Center for Neurogenomics and Cognitive Research, Department of Functional Genomics, Amsterdam Neuroscience, Vrije Universiteit Amsterdam, 1081 Amsterdam, The Netherlands; j.wortel@vu.nl (J.W.); jan.van.weering@vu.nl (J.R.T.v.W.); 3Center for Neurogenomics and Cognitive Research, Department of Clinical Genetics, Amsterdam Neuroscience, Amsterdam UMC location VUmc, 1081 Amsterdam, The Netherlands

**Keywords:** proteomics, mGluR5, mitochondria, mass spectrometry, synapse, CTEP, metabotropic glutamate receptor 5

## Abstract

The metabotropic glutamate receptor 5 (mGluR5) is an essential modulator of synaptic plasticity, learning and memory; whereas in pathological conditions, it is an acknowledged therapeutic target that has been implicated in multiple brain disorders. Despite robust pre-clinical data, mGluR5 antagonists failed in several clinical trials, highlighting the need for a better understanding of the mechanisms underlying mGluR5 function. In this study, we dissected the molecular synaptic modulation mediated by mGluR5 using genetic and pharmacological mouse models to chronically and acutely reduce mGluR5 activity. We found that next to dysregulation of synaptic proteins, the major regulation in protein expression in both models concerned specific processes in mitochondria, such as oxidative phosphorylation. Second, we observed morphological alterations in shape and area of specifically postsynaptic mitochondria in mGluR5 KO synapses using electron microscopy. Third, computational and biochemical assays suggested an increase of mitochondrial function in neurons, with increased level of NADP/H and oxidative damage in mGluR5 KO. Altogether, our observations provide diverse lines of evidence of the modulation of synaptic mitochondrial function by mGluR5. This connection suggests a role for mGluR5 as a mediator between synaptic activity and mitochondrial function, a finding which might be relevant for the improvement of the clinical potential of mGluR5.

## 1. Introduction

The group 1 metabotropic glutamate receptor 5 (mGluR5) is a G-protein coupled receptor expressed widely across the brain, predominantly in hippocampus, striatum and cortex [1]. Pharmacological and genetic animal models have revealed a major role of mGluR5 in learning and memory in hippocampus [2,3,4,5]. mGluR5 is an essential postsynaptic modulator of synaptic plasticity [6,7], facilitating the induction and persistence of long-term potentiation (LTP) and mediating long-term depression (LTD) [2,8,9]. This type of LTD requires the synthesis of new proteins, a process regulated by mGluR5 at multiple levels [10]. Moreover, this receptor is also known to participate in synapse formation, maintenance and maturation [11,12].

mGluR5 regulation of the synapse is mediated by an intricate signal transduction mechanism, which involves the interplay of several signaling pathways (reviewed in [13,14]). The coordinated action of multiple second messengers [such as inositol triphosphate (IP3), Ca2+ and mitochondrial reactive oxygen species (ROS) [15]], protein kinases (e.g., PKC, MAPK, mTOR and ERK1/2) and scaffold protein interactions [16] comprises the cascades underlying mGluR5 functions. This complicated signaling system ultimately leads to the regulation of diverse downstream processes and illustrates the extensive role of mGluR5 in synaptic function at different levels. For instance, mGluR5 is preferentially coupled to Gq/G11 to activate phospholipase C, producing IP3 and releasing Ca2+ from intracellular stores. In the canonical signaling, this leads to the activation of kinases, e.g., CaMKII, PKC, that modulate the function of ion channels, such as the NMDAR, which subsequently affects synaptic plasticity [13]. Alternatively, mGluR5-mediated production of IP3 can also lead to the generation of mitochondrial ROS, which activate ERK and PKA to increase neuronal excitability [15].

mGluR5 has been implicated in multiple brain disorders, including schizophrenia [17], Alzheimer’s disease [18,19], Parkinson’s disease [20], major depressive disorder (review [21]) and fragile X syndrome [22,23,24]. In line with this, the therapeutic potential of mGluR5 pharmacological modulation, in particularly allosteric inhibition, has been extensively described, e.g., in addiction (reviewed in [25,26]), Huntington’s disease [27], chronic stress and depression [28,29], Alzheimer’s disease [30], fragile X syndrome [31] and Parkinson’s disease [32]. Despite robust pre-clinical data, mGluR5 antagonists failed in clinical trials for depression (reviewed in [33]), Parkinson’s disease [34] and fragile X syndrome [35].

The relevance of mGluR5 in synaptic function, various brain disorders and its therapeutic potential have been greatly demonstrated. However, the negative outcome of mGluR5 antagonists in clinical trials reveals the need for a better understanding of the complex processes underlying mGluR5 function and inhibition at the synapse. Here, we used a proteomic approach to investigate the synaptic response to the reduction of mGluR5 activity. By using pharmacological and genetic mouse models, we found that the most prominent effect of mGluR5 inhibition and gene deletion was the change in protein expression of specific pathways in the mitochondria. Electron microscopy analysis revealed the alteration in shape and area of postsynaptic mitochondria in mGluR5 KO synapses, whereas computational and biochemical assays suggested an increase of mitochondrial function in neurons. Together, our observations uniquely indicated the modulation of mitochondrial function by mGluR5, which may be relevant for improving the clinical potential of mGluR5.

## 2. Materials and Methods

### 2.1. Animals and Drug Administration

Grm5 knock-out mice (The Jackson Laboratory, Maine, ME, USA; from strain B6.129–Grm5tm1Rod/J; stock number 003558) were bred and maintained on a C57Bl/6J background in our facility. Male wild type C57Bl/6J mice aged 2-3 months were treated by intraperitoneal (i.e.,) injection of 2 mg/kg 2-chloro-4-((2,5-dimethyl-1-(4-(trifluoromethoxy)phenyl)-1*H*-imidazol-4-yl)ethynyl)pyridine (CTEP, Axon Medchem, Groningen, The Netherlands; cat. number 1972), as previously described [9,30,36], or vehicle (1% DMSO, 5% Tween-80, 30% PEG, 65% saline). Solution was passed through a 0.45 μm filter and adjusted to an administration volume of 10 mL/kg. Animals were sacrificed 2 h or 1 day after treatment. After removing the brains, the hippocampus was dissected and stored at −80 °C until further use. The experiments were performed according to the guidelines approved by the Central Committee for Animal Experiments (CCD) and the Animal Welfare Body (IVD) of the Vrije Universiteit Amsterdam.

### 2.2. Synaptosome Preparation

Synaptosomes were individually prepared from 6 WT and 6 KO mice, as well as 6 WT mice treated with CTEP or vehicle as previously described [37]. Hippocampi were homogenized in HEPES buffer (5 mM HEPES, pH 7.4, 0.32 M sucrose, with EDTA-free protease inhibitor cocktail, Roche, Basel, Switzerland) in a dounce homogenizer (12 strokes, 900 rpm). The homogenate was centrifuged at 1000× *g* for 10 min and the supernatant was subsequently centrifuged in a 0.85/1.2M sucrose gradient at 100,000× *g* for 2 h. The synaptosomes collected from the 0.85/1.2 M sucrose interface were concentrated by centrifugation at 18,000× *g* for 20 min. Protein concentration was determined by using Bradford colorimetric assay (Bio-Rad, Hercules, CA, USA; 5000006EDU).

### 2.3. Sample Preparation for Mass Spectrometry

Synaptosome samples were further processed by filter-aided sample preparation (FASP) with some modifications [38]. In short, 20 μg of protein mixed and incubated with 75 μL 2% SDS, 1 mM Tris(2-carboxyethyl)phosphine for 1 h at 55 °C. Samples were incubated with 0.5 μL 200 mM methyl methanethiosulfonate for 15 min to block the cysteines. Samples were mixed with 200 μL 8 M urea in Tris (pH 8.8) and transferred to Microcon-30 filter tubes (Millipore, Burlington, MA, USA). Next, the samples were centrifuged 15 min at 14,000× *g* and sequentially washed four times with 200 μL 8 M urea in Tris buffer and 4 times with 50 mM ammonium bicarbonate. Protein digestion was performed with 0.7 μg Trypsin/Lys-C Mix (MS grade, Promega; Madison, WI, USA) in 50 mM ammonium bicarbonate overnight at 37 °C. Peptides were recovered with 200 μL 50 mM ammonium bicarbonate, dried in a SpeedVac and stored at −20 °C until use.

### 2.4. LC-MS Analysis

Peptide samples were analyzed by micro LC-MS/MS as described previously [39]. Briefly, we used an Ultimate 3000 LC system (Dionex, Thermo Scientific, Waltham, MA, USA) equipped with a 5 mm Pepmap 100 C18 column (300 μm i.d., 5 μm particle size, Dionex) to trap the peptides and a 200 mm Alltima C18 column (300 μm i.d., 3 μm particle size) for fractionation. A linear gradient of acetonitrile was applied in the mobile phase at a flow rate of 5 μL/min, from 5 to 18% in 88 min, to 25% at 98 min, 40% at 108 min and to 90% in 2 min. Peptides were electro-sprayed with a micro-spray needle (at 5500 V) into a TripleTOF 5600 mass spectrometer (Sciex, Framingham, MA, USA).The data-independent acquisition method consisted of a parent ion scan of 150 ms followed by Sequential Window Acquisition of all THeoretical mass spectra (SWATH) windows of 8 Da with 80 ms scan time, stepped through 450–770 *m*/z mass range. The collision energy for each window was determined for a 2+ ion centered upon the window with 15 eV spread.

Spectronaut Pulsar (v12.0.20491, [40]) was used for data analysis with a spectral library generated from synapse-enriched samples containing spike-in iRT reference peptides (Biognosys, Schlieren, Switzerland). Cross-run normalization was enabled and all other parameters were set at default. The peptides considered for down-stream analysis were quantified with a Q-value ≤ 10^−3^ across all samples in at least half of the groups in each dataset (allowing one outlier within each group). Only proteins with at least two peptides were considered. The mass spectrometry proteomics datasets generated during the current study are available in the PRIDE repository, with the identifier PXD023809.

### 2.5. SDS-PAGE Immunoblotting

Synaptosome samples were mixed with 5× Laemmli buffer and incubated at 90 °C for 5 min. Proteins were separated on SDS-polyacyrlamide gels containing 2,2,2-trichloroethanol for total protein amount visualization. After electrophoresis, gels were scanned using a Gel Doc EZ imager (Bio-Rad) and electro-transferred onto a PVDF membrane overnight at 40 V. Membranes were blocked with 5% non-fat milk (Sigma-Aldrich, St. Louis, MO, USA), incubated with primary antibody at 4 °C for 1 h and then with matching HRP-conjugated secondary antibodies at 4 °C for 1 h (Agilent Dako, Santa Clara, CA, USA). After washing, the membranes were scanned with Femto ECL Substrate (Thermo Fisher Scientific, Waltham, MA, USA) using the Odyssey Fc system (LI-COR Bioscience, Lincoln, NA, USA). Images were quantified using Image Studio software (version 2.0.38). Differences in loading were corrected using the quantification of the total protein amount and immunoblot signals were normalized to the controls. The following primary antibodies were used: anti-ME3 (Abcam, Cambridge, UK; ab172972), anti-mGluR5 (GenScript, Piscataway, NJ, USA; A01493), anti-Rat mGluR1α (BD Biosciences, San Jose, CA, USA; 556331) and anti-PSD-95 (NeuroMab, Davis, CA, USA; 75-028).

### 2.6. Electron Microscopy (EM)

Wild type and mGluR5 KO mice (6 per group) were perfused with ice-cold phosphate-buffer saline (PBS, pH 7.4), followed by 4% paraformaldehyde (PFA) in PBS. Brains were immersed overnight in 4% PFA in PBS and then in 30% sucrose, 0.02% NaN3 in PBS. Brains were sliced in 40 μm sagittal sections with a cryostat. Sections were sequentially post-fixed in 1% osmium tetroxide, 1.5% potassium ferricyanide, dehydrated in an ascending ethanol series, and embedded in epoxy resin. Hippocampal CA1 region was cut into 80 nm sections and contrasted with 0.5% uranyl acetate and lead citrate. Transmission EM analysis was performed at 100,000 magnifications using a JEOL1010 electron microscope (JEOL, Tokyo, Japan) and a side-mounted CCD camera (Morada, EMSIS, Munster, Germany). Digital images were processed with iTEM analysis software (EMSIS).

The synapse length (active zone and postsynaptic density), mitochondrial compartment area, perimeter and distance to the synapse (shortest distance to the active zone or postsynaptic density) were determined in at least 200 synapses per genotype using ImageJ software (imagej.nih.gov/ij, accessec on 23 May 2021). Both symmetric and asymmetric synapses were included in the analysis. Only mitochondria within 500 nm from the center of the active zone or postsynaptic density were considered [41]. The elongation index corresponds to 1/circularity, where circularity was calculated as 4π x Area/Perimeter^2^ [42]. All image acquisition and analyses were performed blinded for genotype.

### 2.7. NADP/H and 2-Thiobarbituric Acid Reactive Substances (TBARS) Assay

Total NADP/H level was determined with a NADP/NADPH assay kit (Abcam, ab65349) following the provided protocol. In brief, 5 WT and 5 mGluR5 KO mouse hippocampi were homogenized in 500 μL extraction buffer and centrifuged at 20.000× *g* for 5 min at 4 °C. The supernatant was transferred to a 10k Da column (Sartorius Stedim Biotech, Gottingen, Germany) and centrifuged at 4000× *g* for deproteinization. The unfiltered protein fraction was kept for protein quantification using Bradford colorimetric assay (Bio-Rad, Hercules, CA, USA, 5000006EDU). Cycling reaction mix was added to each sample in duplicate and developed for 3 h. Samples were scanned (optical density at 450 nm) in a SpectraMax i3 (Molecular Devices, San Jose, CA, USA), individual sample background was subtracted and protein concentration was used for normalization.

Total malonaldehyde (MDA) level was determined with a TBARS assay kit (Oxford Biochemical Research, Rochester Hills, MI, USA; FR45). Hippocampi dissected from 7 WT and 5 mGluR5 KO mice were homogenized, centrifuged and filtered as described above. Samples were mixed in duplicate with indicator solution and incubated at 65 °C for 45 min. Samples were scanned (585 nm emission, 532 nm excitation) in a SpectraMax i3 (Molecular Devices), background was subtracted and protein concentration was used for normalization

### 2.8. Gene Ontology (GO) Enrichment Analysis and Databases

GO enrichment analysis was performed using g:Profiler [43], with g:SCS method for multiple testing correction and all proteins identified as background. Only proteins labeled as “Known mitochondrial” from MitoMinner [44] were considered for down-stream analysis on mitochondrial proteins. Mitochondrial proteins were classified based on their subcellular localization as retrieved from Uniprot, protein complexes from CORUM [45] and functional annotation as curated by MitoXplorer [46]. Proteomic data was visualized onto the electron transport chain (wikipathways WP111_107324, [47]) using PathVisio [48].

### 2.9. Cell-Type Enrichment Analysis

Expression weighted cell-type enrichment (EWCE) analysis was performed as previously described [49]. We used a publicly available single-cell RNA sequencing (scRNA-seq) dataset generated from mouse primary visual cortex [50], including only the exonic data. Uninformative cell-types were excluded. Additionally, cell-type enrichment analysis was performed as implemented in FUMA [51], using 29 of the available pre-processed scRNA-seq datasets generated from nervous system tissue. All mitochondrial proteins identified in the CTEP proteomic dataset were used as background.

### 2.10. Gene Expression Correlation Analysis

Grm5 gene expression was correlated with all mitochondrial and non-mitochondrial genes expression based on a reference scRNA-seq dataset, generated from mouse primary visual cortex [50]. The raw exonic count data was converted into counts per million (CPM) and log10 transformed with pseudo-count 1. Per gene per cell type (subclass) average was calculated for expression values different from 0. Uninformative cell-types were excluded (i.e., “Low quality”, “Batch grouping”, “High intronic”, “Doublet”) and expression data was median normalized. Spearman correlation was performed between Grm5 expression data and the rest of the genome.

### 2.11. Experimental Design and Statistical Analyses

For proteomic analysis, the experiments were performed in randomized blocks [52]. Protein abundances were Loess normalized with Limma R package (‘normalizeCyclicLoess’ function, 10 iterations, ‘fast’ method) [53]. Empirical Bayes moderated t-statistics were applied (*p* < 0.01) using the same R package (‘eBayes’ and ‘topTable’ functions), without multiple testing correction to allow more proteins into the pathway analysis. For EM analysis, outlier identification was performed using GraphPad Prism 6 (ROUT, 1%). Nested statistics were applied by using a linear mixed-effects model as implemented in lme4 R package [54]. Student’s *t* test was applied to analyze the percentage of synapses with mitochondria. For gene expression correlation analysis, the statistical assessment was performed using corrplot R package with ‘fdr’ adjustment for multiple testing correction. Correlations with FDR corrected *p*-values < 0.05 were considered significant. For enrichment analyses, the statistical tests integrated in each method were used. Shapiro-Wilk test and visual inspection were used to test for normality. Other statistical analyses were performed using Student’s *t* test.

## 3. Results

### 3.1. Mitochondrial Protein Expression Is Regulated in mGluR5 KO

To investigate the mechanism behind the function of mGluR5 in the synapse, we first applied quantitative proteomics to wild type (WT) and mGluR5 knockout (KO) mice. The synaptic proteome was enriched by preparing synaptosomal fractions from mouse hippocampus. Proteins were digested into peptides and analyzed by mass spectrometry using the label-free Sequential Window Acquisition of all THeoretical mass spectra (SWATH) acquisition method [38]. Data analysis was performed considering only proteins with >2 high quality peptides for downstream analysis. After filtering, 11,400 peptides from 1766 proteins were identified (Figure 1A, Appendix A). Proteins were quantified with a median coefficient of variation of 9% and 8% for the 6 biological replicates of WT and KO synaptosome preparations, respectively (Appendix A). As a result, the expression of 26 proteins was found increased and 38 decreased in mGluR5 KO synaptosomes (eBayes *p* ≤ 0.01, Figure 1A).

To gain insight in the underlying processes regulated, we performed gene ontology enrichment analysis (Figure 1B, Appendix A). The most prominent overrepresentation identified was the mitochondria, with general ontology terms such as ‘mitochondrion’, and more specifically the mitochondrial inner membrane. In total 73% of all up-regulated proteins were mitochondrial. Moreover, several of these proteins also showed high fold-changes, such as Me3 (malic enzyme 3), which is the most up-regulated protein with ~2.5 times higher protein expression in mGluR5 KO than in WT controls (*p* = 6.6 × 10−11). Eighteen proteins from ‘metabolic pathways’ (KEGG), mainly related to the glucose metabolism, were found consistently down-regulated in mGluR5 KOs (0.85-fold of the WT protein expression on average). Other metabolic pathways were obtained from other sources, such as ‘small molecule biosynthetic process’ (Gene Ontology biological process), ‘metabolism of carbohydrates’ (Reactome), ‘glycolysis and gluconeogenesis’ (Wikipathways).

Several known synaptic proteins, as annotated in SynGO [55], were found among the regulated proteins. As expected, mGluR5 (Grm5) was the protein most significantly decreased in expression; compared to the background signal of the KO where Grm5 was not identified. Other synaptic proteins with decreased expression level included mGluR1 (Grm1), Gad1 and Pacsin1; while the expression of PrkCb (Protein kinase C beta type), Gpc1 and Kcnd3 (Potassium voltage-gated channel subfamily D member 3) were found increased, which may involve compensatory mechanisms to the lack of mGluR5 activity.

As an independent validation of the results obtained by mass spectrometry, we performed immunoblotting analysis of several proteins and with different directions of regulation (Figure 1B). In all cases we found good agreement between approaches, i.e., mGluR5 was absent in KO mice, mGluR1 was significantly down-regulated, mitochondrial Me3 was strongly up-regulated, and Dlg4 (PSD95) was not found regulated with either method. Taken together, these data revealed a predominant increase of mitochondrial proteins expression in mGluR5 KO synaptosomes and, to lesser extent, the decrease of metabolic proteins expression and differential expression of synaptic proteins.

### 3.2. Postsynaptic Mitochondrial Morphology Is Altered in mGluR5 KO

We next asked whether the mitochondrial protein differential expression in mGluR5 WT/KO may be derived from changes in the structure and/or trafficking of mitochondria to the synapse. To answer this, we used transmission electron microscopy (EM) of hippocampal CA1 region synapses from WT and mGluR5 KO brains sections (Figure 2A). First, 209 WT and 247 KO synapses were analyzed to assess whether there is a preferential mitochondrial localization to the synapse in mGluR5 KO. No significant difference was observed in the percentage of synapses containing mitochondria (36% of the WT and 42% of the KO synapses; Figure 2B). In addition, the distance of the mitochondria to the synapse (active zone or postsynaptic density) was not found significantly altered (Figure 2C), neither for presynaptic and postsynaptic mitochondria separately (Appendix A). Thus, mitochondrial localization does not seem to be altered in mGluR5 KO synapses.

Second, the morphology of the synaptic mitochondria was measured. We analyzed the area and perimeter of 59 WT and 86 KO synaptic mitochondrial compartments. Both parameters were found significantly decreased in mGluR5 KO synapses (Figure 2D), while the synaptic active zone length did not change (Appendix A). Furthermore, we found the postsynaptic mitochondrial area and perimeter significantly decreased in mGluR5 KOs (Figure 2E), whereas the presynaptic mitochondria were not affected (Figure 2F).

Next, we calculated the elongation index, which gives an indication of the shape of the mitochondria. A perfect circular shape corresponds to an elongation index of 1, while higher values indicate more elongated or complex shapes. The elongation index of postsynaptic mitochondria was found reduced in mGluR5 KOs (Figure 2E), whereas for presynaptic mitochondria this was not significantly different from WTs (Figure 2F). Altogether, the mitochondrial compartment in mGluR5 KO exhibited a smaller area, perimeter and elongation index for the postsynapse, but not for the presynapse, matching the mainly postsynaptic expression of mGluR5.

### 3.3. Acute Pharmacological Inhibition of mGluR5 Regulates Mitochondrial Protein Expression

To explore the synaptic mechanisms underlying mGluR5 function in an alternative model of diminished mGluR5 activity, we investigated acute pharmacological inhibition of mGluR5 using quantitative proteomics. Adult WT mice were treated once or twice with CTEP, a negative allosteric modulator of mGluR5 with a long half-live in mice [31,36]. Animals were sacrificed 2 (CTEP2h), 24 (CTEP1d) and 48 h (CTEP2d) after the first administration (Figure 3A). Hippocampal synaptosomes were isolated and analyzed by mass spectrometry as indicated above, using mice treated with vehicle as control. From the 2072 proteins quantified across all experimental conditions (Appendix A), in total 52 proteins were significantly regulated upon CTEP treatments (eBayes *p* ≤ 0.01, Figure 3B,C); 45 were found regulated in the CTEP2h group (Figure 3B), while only five proteins were regulated in CTEP1d (Figure 3C) and three in CTEP2d (Figure 3D). All three groups were quantified with a low 8% median coefficient of variation within replicates (Appendix A).

Next, we performed gene ontology enrichment analysis to functionally annotate the regulated proteins upon CTEP treatment (Figure 3E, Appendix A). Interestingly, 79% of the proteins found up-regulated belong to the mitochondrion, which parallels the results obtained from mGluR5 KOs. Whereas little overlap was found for the proteins significantly regulated in mGluR5 KO and CTEP-treated mice (Appendix A), gene ontology enrichment analysis revealed the same 8 GO cellular component terms in both models, including mitochondrion, mitochondrial envelope and mitochondrial inner membrane. Furthermore, we observed that the mitochondrial and metabolic proteins regulated in mGluR5 KOs followed the same direction of regulation in CTEP2h with smaller fold-change (Appendix A). Among the down-regulated proteins, we found the synaptic Bin1-Dnm1-Ehbp1 complex, Arhgap44, Map2k1 and Pak3, as well as Pacsin1 and Got1, of which expression was also significantly decreased in the mGluR5 KO. All in all, the expression of multiple mitochondrial proteins was found increased 2 h after the CTEP treatment, partly replicating the effects identified in the KO.

### 3.4. Reduced mGluR5 Activity Alters Specific Mitochondrial Pathways

To gain insight in the mGluR5-mediated mitochondrial protein regulation, we inspected several aspects of the mitochondrial proteome. First, we explored the distribution of expression fold-changes in CTEP2h for all mitochondrial proteins, as annotated by MitoMiner (Figure 4A). We found that mitochondrial proteins followed a significant bimodal distribution compared to a randomized protein selection (*p* < 2.2 × 10^−16^, 100 iterations), i.e., mitochondrial proteins tended to form two groups in the CTEP2h treatment, one without expression changes and one with higher expression compared to controls. To uncover the specific mitochondrial processes implicated, the mitochondrial proteins were grouped based on their cellular compartment, protein complex and biological functions, as retrieved from Uniprot, CORUM and MitoXplorer, respectively. Regarding the distribution over the sub-organelle compartments, proteins located in the mitochondrial inner membrane and intermembrane space showed higher fold-changes, compared with the mitochondrial matrix (Appendix A). Regarding the biological function, the highest fold-changes were found for proteins related to the oxidative phosphorylation, mitochondrial carrier and dynamics (Figure 4B). Accordingly, the protein complexes with highest fold-changes correspond to oxidative phosphorylation and mitochondrial intermembrane space bridging complexes (Appendix A). The visualization of the oxidative phosphorylation pathway confirmed a broad coverage of proteins identified, with a 1.15-fold median increased expression in synaptosomes of CTEP-treated mice (Figure 4C). In contrast, the cytosolic proteins from the pentose phosphate pathway, glycolysis and ROS defense showed the lowest fold-changes, in line with the decreased protein expression of metabolic pathways observed in the mGluR5 KO. Altogether, the short-term inhibition of mGluR5 appears to affect the protein expression of specific pathways of the mitochondrial proteome, in particular the oxidative phosphorylation.

To understand the functional implication of the mitochondrial pathways regulated, the levels of NADP/H (a cofactor of several mitochondrial enzymes) and oxidative damage (partly derived from the oxidative phosphorylation activity) were analyzed in mGluR5 KO hippocampus. Total NADP/H in KO hippocampus was 21% higher in comparison to WT (Figure 4D), which matches well with the increased protein expression of NAD(P)H-dependent mitochondrial enzymes, such as Me3. To assess oxidative damage, we measured the malonaldehyde (MDA) produced via lipid peroxidation as a consequence of oxidative stress. Total MDA level in KO hippocampi was significantly increased 38% compared to WTs (Figure 4E), which indicates elevated lipid damage resulting from oxidative stress. Taken together, the increased level of NADP/H and oxidative damage observed in mGluR5 KO hippocampus suggests higher oxidative stress partly derived from the augmented mitochondrial activity, including the oxidative phosphorylation.

### 3.5. Neuronal Mitochondrial Proteins Are Mostly Regulated by the mGluR5 Inhibition

In the adult mouse brain, mGluR5 is mostly expressed in neurons [56,57], however, other cell types may also be indirectly implicated in the mGluR5-mediated mitochondrial regulation. While the synaptosome preparations analyzed mostly contain neuronal synaptic mitochondria, impurities of free mitochondria from other cell-types might also be present in this fraction. Interestingly, the expression of mitochondrial proteins differs across tissues and cell types to adjust to different tasks and demands [58,59] and, thus, these expression differences can be used to discriminate between cell-types. To determine the cells most affected by the manipulation of mGluR5 function, we examined the cell-type distribution of the mitochondrial proteome.

First, we performed expression weighted cell-type enrichment (EWCE) analysis on the mitochondrial proteins significantly regulated by CTEP2h treatment, with all identified mitochondrial proteins as background. When using a single-cell transcriptomic (scRNA-seq) dataset from mouse cortex [50], the most significant cell-types obtained correspond to excitatory neurons, including neuronal types from multiple cortical layers (Figure 4F, Appendix A). To confirm this result using independent single-cell datasets, we performed cell-type enrichment analysis using 29 publicly available scRNA-seq datasets from mouse and human nervous system tissue as implemented in FUMA [51]. Accordingly, the vast majority of the most significant cell-types obtained correspond to neurons, including 84% of the three most significant cell-types per dataset (Appendix A, Appendix A).

To test whether mitochondrial proteins with stronger CTEP-induced regulations are associated with a specific cell-type, we performed EWCE analysis on the 50 mitochondrial proteins with higher fold-change upon CTEP2h treatment. Similar to the significantly regulated proteins, higher fold-change mitochondrial proteins were enriched in neuronal cell-types (Figure 4F, Appendix A). As an independent assessment, we correlated the genetically determined expression of all mitochondrial and non-mitochondrial genes with mGluR5 (Appendix A). mGluR5 mostly exhibited either a positive or negative correlation with mitochondrial genes, in contrast with the absence of correlation observed between mGluR5 and all non-mitochondrial genes. The expression profiles of mitochondrial genes significantly correlated with mGluR5 revealed that the positive and negative correlations correspond to genes little and highly expressed in non-neuronal cell types, respectively (Appendix A). Moreover, positively correlated genes showed a higher fold-change in protein expression than the negatively correlated genes upon CTEP treatment (Appendix A). The same trend was found using single-cell protein expression [59], where mitochondrial marker proteins for granule cells showed higher fold-changes than for astrocytes (Appendix A). Taken together, the mitochondrial proteins most regulated by the treatment with CTEP appear to be enriched in neuronal cell-types, matching the mostly neuronal mGluR5 expression, while non-neuronal mitochondrial proteins were less affected.

## 4. Discussion

In this study, we investigated synapse adaptation to loss or impaired mGluR5 function from a proteomic perspective. By using a mGluR5 KO mouse model, we first assessed the adaptation of synaptic proteins to the chronic lack of mGluR5 function throughout development and lifespan. Pharmacological inhibition of mGluR5 in vivo allowed us to evaluate the acute synaptic modulation derived from the reduction of mGluR5 activity in adult mice. Despite fundamental differences between these models, we found that the major synaptic protein expression changes in both models concerned mitochondrial functions. This mGluR5-dependent mitochondrial modulation was further characterized by electron microscopy, biochemical and computational analysis.

Mass spectrometry analysis of mGluR5 KO synaptosomes led to the identification of 64 significantly regulated proteins from different functional categories. Among the synaptic proteins, mGluR1 (Grm1) and PKC beta (Prkcb) expression was found decreased and increased, respectively. mGluR1 and mGluR5 interact in a protein complex and constitute the group 1 metabotropic glutamate receptors family [60], while PKC is involved in mGluR5 signaling pathway. The increased PKC beta expression and co-regulation of mGluR1-mGluR5 suggest the rearrangement of downstream effectors rather than the functional compensation between receptor family members for the lack of mGluR5 signaling. Additionally, Pacsin1 expression was reduced in both the mGluR5 KO and CTEP-treated mouse synaptosomes, a protein known to promote AMPA and NMDA receptor endocytosis and trafficking [61,62,63]. Moreover, Arhgap44 [64], Map2k1 [65] and Pak3 [66] were found down-regulated in CTEP-treated mice, which are involved in AMPA receptor trafficking and recycling. Given the role of mGluR5 in the modulation of AMPA receptor and induction of NMDA receptor trafficking to the surface [67], the action of Pacsin1, Arhgap44, Map2k1, Pak3 and mGluR5 is likely coordinated to regulate the overall synaptic response to glutamate. Taken together, mGluR5 KO and CTEP-treated mice showed the expression regulation of multiple synaptic proteins known to be functionally related to mGluR5.

Despite the regulation of synaptic proteins, the most prominent effect observed in both models was the increased expression of mitochondrial proteins. From all up-regulated synaptosome proteins, 73% and 79% were mitochondrial for mGluR5 KO and CTEP-treated mice, respectively. Mitochondrial and synaptic function are tightly interconnected [68]. Neuronal activity triggers a rapid burst of mitochondrial fission and elevation of mitochondrial calcium [69]. Vice versa, mitochondria located in axons and dendrites are essential for synaptic transmission and plasticity via different mechanisms, such as the regulation of calcium homeostasis, ATP production and protein translation [70,71]. Since mGluR5 is critically implicated in protein synthesis and LTD, a functional link between mitochondria and mGluR5 is plausible. Interestingly, mitochondrial ROS have been previously shown to act as second messenger in mGluR5 signaling pathway [15,72]. Moreover, several lines of evidence suggest the essential role of ROS, partly produced by mitochondria, in physiological synaptic plasticity, learning and memory, while the pathological disruption of ROS balance impairs these processes [73]. Considering the common role of mitochondria and mGlulR5 as modulators of synaptic plasticity, mGluR5 may represent a key hub for the crosstalk between synaptic transmission and mitochondria.

In addition, several proteins involved in glucose metabolism were found down-regulated in mGluR5 KOs, but not as extensively in CTEP-treated mice. Although a relatively low overlap in protein identity of significant proteins was found between both models, the metabolic and mitochondrial proteins regulated in mGluR5 KOs showed the same trend upon CTEP treatment with smaller effect size. This observation suggests that the acute inhibition of mGluR5 can modulate the levels of the same proteins at the synapse, but not as strong as the chronic lack of mGluR5 activity in the KOs. Remarkably, the effect of the pharmacological inhibition of mGluR5 seems quickly reversible since it was not found 24 h after CTEP administration, regardless of the long CTEP half-life [36]. Together, acute and chronic mGluR5 function inhibition both induce a substantial increase in mitochondrial protein expression.

Observations combined from mGluR5 KO and CTEP-treated mice revealed several aspects of the mGluR5-mediated mitochondrial regulation. First, no preferential synaptic localization was found for mitochondria in the mGluR5 KO, suggesting that the increased protein expression was not simply caused by a higher number of mitochondria included into the synaptosome preparation. Accordingly, pharmacological inhibition of mGluR5 seems to regulate the protein expression of specific pathways rather than the whole mitochondrial proteome. Second, a reduced area, perimeter and elongation index of synaptic mitochondria were found in mGluR5 KOs while the synapse length was unaffected, implicating mitochondria with smaller size and more circular shape. Since mitochondrial morphology and function are closely associated, these morphological changes might indicate a modulation of mitochondrial activity. Indeed, pathway analysis from CTEP-induced mice indicated regulation of oxidative phosphorylation proteins, which are implicated in the production of oxidative stress and ROS. Consistently, an increased oxidative damage and NADP/H, cofactor of multiple mitochondrial enzymes, were observed in mGluR5 KO, possibly underlying a higher mitochondrial activity. Mitochondrial morphology and activity can be adjusted by synaptic activity via fusion/fission events, which are required for synaptic plasticity [69]. Considering the function of mGluR5 as a glutamate receptor, these observations suggest that mGluR5 might be implicated in the connection between synaptic activity, the release of glutamate and mitochondrial function. Lastly, the morphological alteration of mitochondria were found only in the postsynaptic compartment, matching the main subcellular localization of mGluR5 [1]. In parallel, cell-type enrichment analysis showed that the mitochondrial pathways mostly regulated by CTEP-treatment correlated best with neuronal cell types, consistently with the primary neuronal mGluR5 expression in adult brain [56,57]. Collectively, these results strongly suggest a direct link between mGluR5 and the mitochondrial function in the synapse.

In previous studies, indications of the association between mGluR5 and mitochondrial function have been found. MGluR5 was shown to modulate ATP levels in different types of cells [74,75,76], production of ROS [15,72,75,77], mitochondrial mass [75] and mitochondrial biogenesis [78]. Our study revealed that mGluR5, in addition to these, can also regulate specific mitochondrial proteins, morphology and NADP/H level. Our observation of an mGluR5-dependent postsynaptic mitochondrial morphology may partly explain the dysfunctional mitochondrial respiration and morphology recently reported in a fragile X syndrome mouse model [79], in which mGluR5 plays a predominant role [80]. Moreover, the proteomic data provides a new entry into exploration of the specific mitochondrial pathways implicated and the mechanisms by which mitochondrial function responds to mGluR5. Among the multiple downstream effectors of mGluR5 signaling, PKC beta (Prkcb) level was found increased in mGluR5 KO, and MEK1 (Map2k1) level decreased in CTEP-treated mice. These observations suggest the involvement of Gq-IP3-PKC rather than MEK-ERK signaling pathway in the mGluR5-mediated mitochondrial regulation, which might be considered for further investigation. Accordingly, mitochondrial ROS have been found activated by mGluR5 through IP3 [15], and mitochondrial respiration in synaptic fractions is known to be altered via PKC and IP3 in ischemia [81,82,83]. Taken together, we found new evidence for the connection between mGluR5 and mitochondrial function and provided data to generate hypotheses for future investigation regarding the underlying mechanistic link between mGluR5 and mitochondria, as well as the functional and therapeutic implication of this relation.

Our proteomic analysis indicated an increase in the level of mitochondrial proteins, in contrast with the reduced mitochondrial size shown by EM analysis. Although these results appear to mismatch, both observations could be compatible. The increase in levels of mitochondrial proteins might reflect a different metabolic state of the mitochondria rather than a change in their size. Accordingly, we found only protein levels within specific mitochondrial pathways increased, instead of the whole mitochondrial proteome, and increased NADP/H levels suggesting changes in mitochondrial physiology. Alternatively, the increased protein levels may also be derived from presynaptic mitochondria, for which a morphological alteration was not observed, and/or different types of dendritic spines not easily identified by EM analysis.

In conclusion, the most prominent mGluR5-mediated synaptic response identified was the regulation of mitochondria. Given the important role of mitochondria in disorders of the central nervous system [84] and the broad therapeutic potential of mGluR5, this functional relationship represents an interesting connection to consider regarding improvement of treatments and the outcome of clinical trials. Moreover, we postulate the role of mGluR5 as a regulator of the link between synaptic activity and mitochondrial function.

## Figures and Tables

**Figure 1 cells-10-01375-f001:**
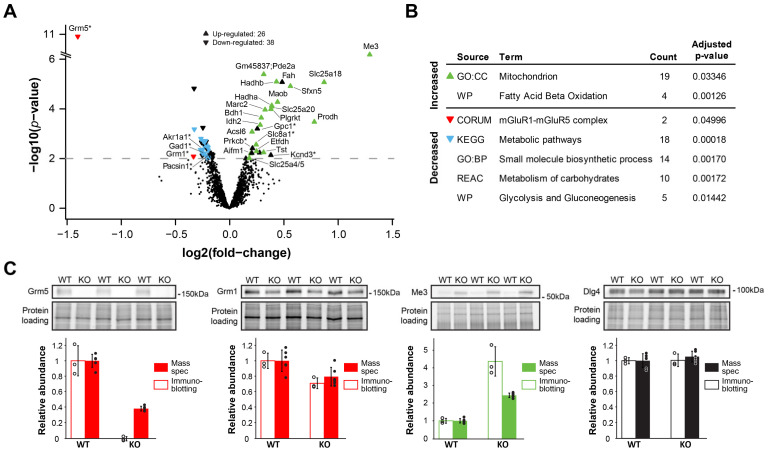
Quantitative proteomics analysis of mGluR5 KO synaptosomes. (**A**) Volcano plot showing the distribution of fold-change protein expression in mGluR5 KO synaptosomes compared to WT mice (N = 6). Proteins above the dotted line were considered significantly regulated (eBayes *p* < 0.01). Proteins are labeled and colored based on the 3 non-overlapping annotations with highest protein count in (**B**). Asterisks in the protein name tags indicate synaptic proteins based on SynGO. (**B**) Gene ontology enrichment analysis for the significantly regulated proteins (*p* < 0.01) depicted in (**A**). The ontology terms with the highest protein count from each database source are indicated. (**C**) SDS-PAGE immunoblot validation of selected proteins found decreased (Grm5, Grm1), increased (Me3), or non-affected (Dlg4) in mGluR5 KO (N = 3). In agreement with the mass spectrometry data, Grm5, Grm1 and Me3 protein expression were found significantly regulated (*t*-test *p* = 0.001, 0.014 and 0.002, respectively), but not Dlg4 (*p* = 0.93). Protein loading was visualized and quantified in-gel using 2,2,2-Trichloroethanol to correct total protein amount differences. CORUM, the comprehensive recourse of mammalian protein complexes; GO:BP, Gene Ontology Biological Processes; GO:CC, Gene Ontology Cellular Compartment; KEGG, Kyoto Encyclopedia of Genes and Genomes; REAC, Reactome; WP, WikiPathways Ontology. All bar graphs, means ± SD.

**Figure 2 cells-10-01375-f002:**
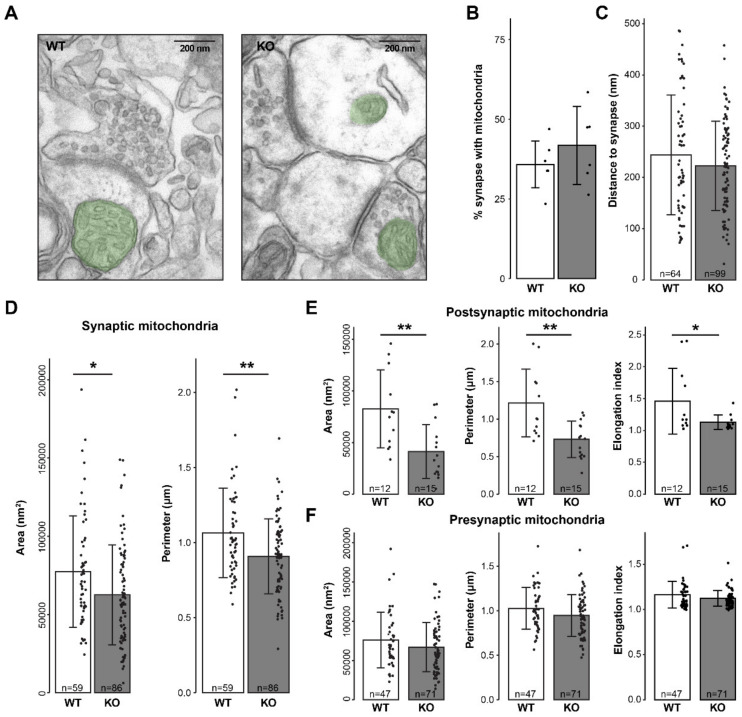
Electron microscopy analysis of mGluR5 KO synaptic mitochondria. (**A**) Representative images of typical asymmetric glutamatergic synapses from WT and mGluR5 KO hippocampus (N = 6). Synaptic mitochondria are highlighted in green. (**B**) Percentage of synapses containing mitochondria in either the pre- or post-synaptic compartment (N = 6, n = 209 for WT; N = 6, n = 247 for KO). No significant difference was found between WT and KO (*t*-test, *p* = 0.64). (**C**) Linear distance from the presynaptic and postsynaptic mitochondria to the active zone and postsynaptic density, respectively. No significant difference was found between WT and KO (nested *p* = 0.29). (**D**) Area and perimeter of mitochondria located in both the pre- and post-synaptic compartment were significantly reduced in KO mice (nested *p* = 0.028 and 0.003, respectively). (**E**) Area, perimeter and elongation index of mitochondria located in the post-synaptic compartment were significantly reduced in KO mice (nested *p* = 0.002, 0.001 and 0.019, respectively). (**F**) No significant differences were found between WT and KO in the area, perimeter and elongation index of mitochondria located in the presynaptic compartment (nested *p* = 0.20, 0.09 and 0.11, respectively). * *p* < 0.05, ** *p* < 0.01. All bar graphs, means ± SD.

**Figure 3 cells-10-01375-f003:**
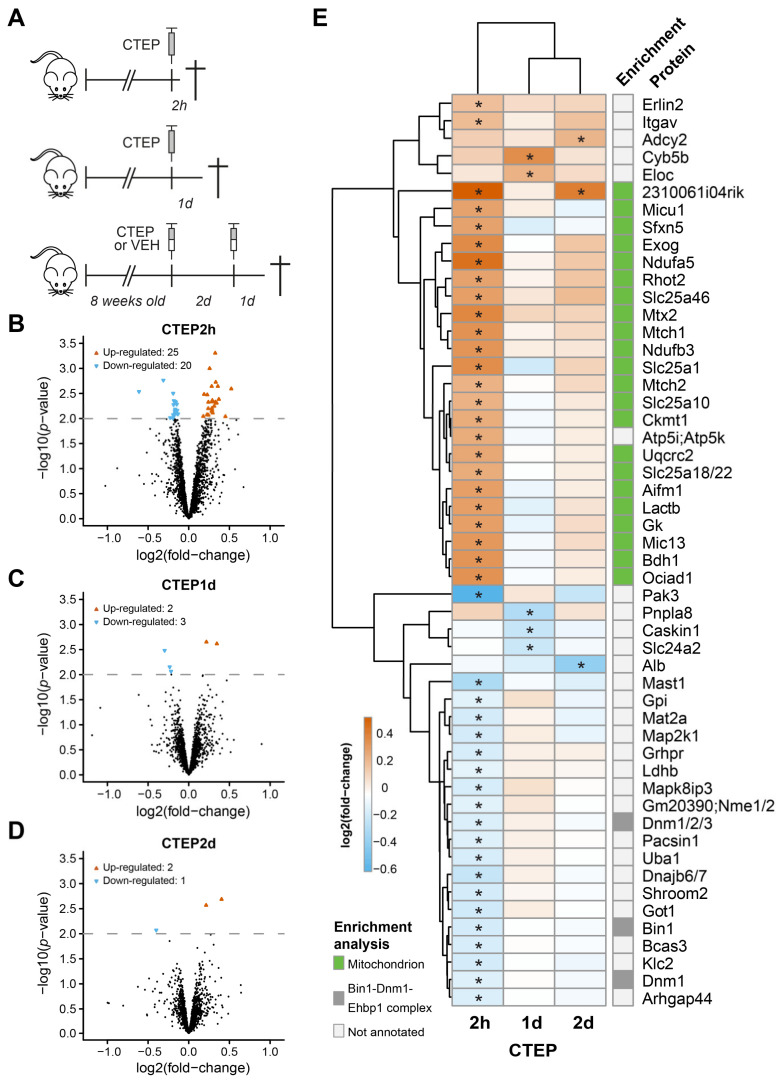
Quantitative proteomics analysis of synaptosomes of CTEP-treated mice. (**A**) Experimental design. Adult WT mice were treated with CTEP or vehicle and sacrificed 2 (CTEP2h), 24 (CTEP1d) and 48 h (CTEP2d) after the first administration as indicated (N = 6). (**B**–**D**) Volcano plots showing the distribution of fold-change protein expression in CTEP-treated mice synaptosomes from the CTEP2h (**B**), CTEP1d (**C**) and CTEP2d (**D**) groups compared to vehicle control. Proteins above the dotted line were considered significantly regulated (eBayes *p* < 0.01). (**E**) Heatmap showing the differentially abundant synaptosome proteins between CTEP-treated and control mice. Color gradient (blue-red) indicates the fold changes compared to vehicle control. Asterisks indicate the experimental group in which the protein was found significantly regulated. Proteins were annotated according to gene ontology enrichment analysis.

**Figure 4 cells-10-01375-f004:**
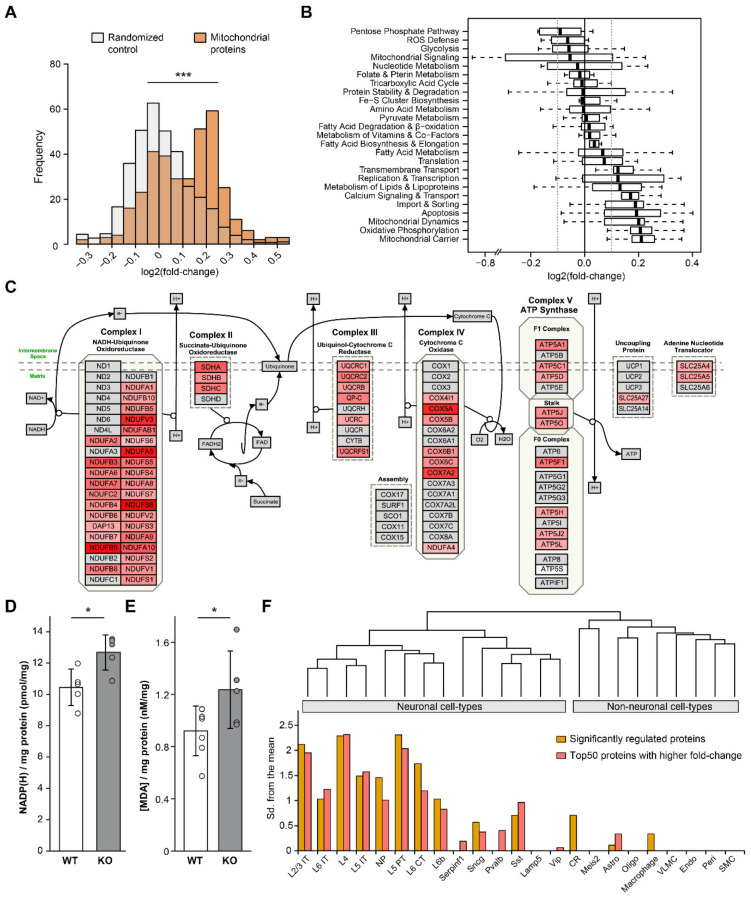
Mitochondrial pathway and cell-type enrichment analysis of CTEP-induced protein expression regulation. (**A**) Distribution of protein expression fold-changes induced by the treatment with CTEP (CTEP2h group) for all quantified mitochondrial proteins and a randomized control (100 iterations). The mitochondrial proteins distribution was significantly different from the control. (**B**) Boxplot showing the distribution of mitochondrial protein expression fold-changes (CTEP2h group) in different functional categories based on MitoXplorer. Solid and dotted vertical lines indicate no change and ±0.1 fold-change (log2), respectively. (**C**) Visualization of CTEP-mediated protein expression modulation onto the mitochondrial electron transport chain (wikipathways WP111_107324). Color gradient represents the protein expression fold-change (CTEP2h group). Proteins in gray were not identified. (**D**) Total level of the cofactor NADP/H was increased in mGluR5 KO hippocampus compared to WT (N = 5). (**E**) Total MDA (malonaldehyde) level was increased in mGluR5 KO hippocampus compared to WT (N = 7 for WT; N = 5 for KO), as an assessment for oxidative damage. (**F**) Expression weighted cell-type enrichment analysis for the differentially abundant mitochondrial proteins and the 50 mitochondrial proteins with larger fold-changes upon CTEP2h treatment. The dendogram represents the hierarchical relationship between cell-types. * *p* < 0.05, *** *p* < 0.001. All bar graphs, means ± SD.

## Data Availability

The mass spectrometry proteomics datasets generated during the current study are available in the PRIDE repository, with the identifier PXD023809.

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
