# Peer review of "Reduced mGluR5 Activity Modulates Mitochondrial Function"

_cells, 2021, doi:10.3390/cells10061375_

Round 1

Reviewer 1 Report

The paper 'Reduced mGluR5 activity modulates mitochondrial function in the synapse', Gonzalez-Lozano et al., presents an extensive proteomic analysis from synaptosomes isolated from control and mGlu5 KO mouse hippocampi. They support their findings with immunoblotting for select up- and down-regulated proteins. Interestingly, mitochondrial genes seem to be the most strongly affected by KO of mGlu5, The authors then perform an EM analysis of synapses and find no differences in mitochondrial localization but that mitochondrial size is reduced in postsynaptic compartments. They then analyze hippocampal synaptosomes from CTEP treated mice and finally test for some particular markers of oxidative phosphorylation, namely NADPH and MDA.

The findings are interesting and the paper is well-written. As a general comment, readability could still be improved if the authors more often stated in which direction effects were seen when citing the literature and used less often only words like 'affects, regulates, participates' in very non-specific ways. For example line 472 'Neuronal activity regulates mitochondrial morphology, function and the physical proximity... would be more informative if it stated 'how' these aspects are regulated i.e. 'Strong neuronal activity causes mitochondrial fission and increases the distance to synapses'.

It is mentioned that the EM analysis was performed blind to genotype, did the blinding include collecting the images or was only the analysis performed blind. Was there blinding used for any of the proteomic analysis?

In the EM section was the analysis restricted to asymmetric synapses? it is not mentioned if all types of synapses were grouped together. It seems a bit counter-intuitive that an upregulation of many mitochondrial genes would be associated with a reduction in size. It would be nice in the discussion to read how the authors interpret this and if there is any precedent in the literature. Did the authors observe any changes in synapse structure/density itself in their data set? Possibly there is a change in the density of synapses that accounts for the lack of effect on number of synapses with mitochondria.

Author Response

The paper 'Reduced mGluR5 activity modulates mitochondrial function in the synapse', Gonzalez-Lozano et al., presents an extensive proteomic analysis from synaptosomes isolated from control and mGlu5 KO mouse hippocampi. They support their findings with immunoblotting for select up- and down-regulated proteins. Interestingly, mitochondrial genes seem to be the most strongly affected by KO of mGlu5, The authors then perform an EM analysis of synapses and find no differences in mitochondrial localization but that mitochondrial size is reduced in postsynaptic compartments. They then analyze hippocampal synaptosomes from CTEP treated mice and finally test for some particular markers of oxidative phosphorylation, namely NADPH and MDA.

The findings are interesting and the paper is well-written. As a general comment, readability could still be improved if the authors more often stated in which direction effects were seen when citing the literature and used less often only words like 'affects, regulates, participates' in very non-specific ways. For example line 472 'Neuronal activity regulates mitochondrial morphology, function and the physical proximity... would be more informative if it stated 'how' these aspects are regulated i.e. 'Strong neuronal activity causes mitochondrial fission and increases the distance to synapses'.

We appreciate the interest and suggestions of the reviewer regarding the readability of our manuscript. We followed the reviewer’s advice and we now included more specific statements throughout the manuscript. This includes changing the term “regulate” for “promote endocytosis” (line 472), “modulation” for “induction” (line 476) and “regulate” for “trigger a rapid burst” (line 485), among others. Also see textual changes in the main text of the manuscript.

It is mentioned that the EM analysis was performed blind to genotype, did the blinding include collecting the images or was only the analysis performed blind. Was there blinding used for any of the proteomic analysis?

Both the image acquisition and analysis were performed blinded for genotype. The main text has been updated to clarify this aspect (page 4 line 170), see highlighted main text. Regarding the proteomic analysis, a “randomized block” experimental design (Festing, M.F.W. (2020). Sci Rep 10, 17577) was followed for the synaptosome isolation and mass spectrometry analysis. To avoid bias, the experiments were performed in blocks consisting of one sample from each group, which provides a good control over the environmental variation. We now included this information on page 5, line 215, see highlighted text.

In the EM section was the analysis restricted to asymmetric synapses? it is not mentioned if all types of synapses were grouped together. It seems a bit counter-intuitive that an upregulation of many mitochondrial genes would be associated with a reduction in size. It would be nice in the discussion to read how the authors interpret this and if there is any precedent in the literature. Did the authors observe any changes in synapse structure/density itself in their data set? Possibly there is a change in the density of synapses that accounts for the lack of effect on number of synapses with mitochondria.

Both symmetric and asymmetric synapses were considered for the analysis. This is now indicated on page 4, line 166. Regarding the synapse structure and density, no significant difference was found in the active zone length between WT and mGluR5 KO synapses (Fig. S2B). Dendritic spine density has been shown previously to be increased in mGluR5 KO mice cortex (Chen, C.C. et al (2012). Neuroscience letters, 524(1), 65–68) and in the prefrontal cortex of rats treated with fenobam, a negative allosteric modulator of mGluR5 (Chen, C.C. et al (2012). Neuroscience letters, 524(1), 65–68). As the reviewer suggested, this could account for the lack of effect on number of synapses with mitochondria and the counter-intuitive reduction in mitochondrial size and upregulation of mitochondrial proteins. Following the recommendation of the reviewer, we now discussed the topic on page 14, lines 557-566, see highlighted text, as: “Our proteomic analysis indicated an increase in the level of mitochondrial proteins, in contrast with the reduced mitochondrial size shown by EM analysis. Although these results appear to mismatch, both observations could be compatible. The increase in levels of mitochondrial proteins might reflect a different metabolic state of the mitochondria rather than a change in their size. Accordingly, we found only protein levels within specific mitochondrial pathways increased, instead of the whole mitochondrial proteome, and increased NADP/H levels suggesting changes in mitochondrial physiology. Alternatively, the increased protein levels may also be derived from presynaptic mitochondria, for which a morphological alteration was not observed, and/or different types of dendritic spines not easily identified by EM analysis.”.

Reviewer 2 Report

The authors used mGluR5 KO and CTEP-treated mouse models and performed a study of synaptosomes at protein, gene, and morphology levels. In mGluR5 KO mice, the authors observed the protein expression changes associated with mitochondria especially oxidative phosphorylation. Postsynaptic mitochondria exhibit smaller areas and perimeters. Similar proteomic changes were also found in the CTEP-treated mice. The results indicate mGluR5 correlated with the regulation of mitochondria and thus important in CNS disorders like oxidative stress.  The strength of this study is providing detailed proteomic and genomic evidence showing mitochondria changes in both mGluR5 KO and CTEP-treated mice.  

The novelty was not highlighted in this paper. The authors should compare with the previous finding and point out the creativities of this study.  As authors mentioned in the section (line 468),  mGluR5 has been known correlated with mitochondrial functions (Li, Ji et al. 2011). Ischemia or oxidative stress have been known to cause PKC changes (Harada, Maekawa et al. 1999, Dave, DeFazio et al. 2008) and synaptosomal changes via IP3 pathways (Wikiel and Strosznajder 1987).  Besides the evidence collected with the new technique, what new information does this study provide?  

Line 350: The subtitle of this section is about “pharmacological” inhibition of mGluR5, however, the results of mGluR5 KO model were also described.

Author Response

The authors used mGluR5 KO and CTEP-treated mouse models and performed a study of synaptosomes at protein, gene, and morphology levels. In mGluR5 KO mice, the authors observed the protein expression changes associated with mitochondria especially oxidative phosphorylation. Postsynaptic mitochondria exhibit smaller areas and perimeters. Similar proteomic changes were also found in the CTEP-treated mice. The results indicate mGluR5 correlated with the regulation of mitochondria and thus important in CNS disorders like oxidative stress.  The strength of this study is providing detailed proteomic and genomic evidence showing mitochondria changes in both mGluR5 KO and CTEP-treated mice.  

The novelty was not highlighted in this paper. The authors should compare with the previous finding and point out the creativities of this study.  As authors mentioned in the section (line 468),  mGluR5 has been known correlated with mitochondrial functions (Li, Ji et al. 2011). Ischemia or oxidative stress have been known to cause PKC changes (Harada, Maekawa et al. 1999, Dave, DeFazio et al. 2008) and synaptosomal changes via IP3 pathways (Wikiel and Strosznajder 1987).  Besides the evidence collected with the new technique, what new information does this study provide?

We sincerely thank the reviewer for the recommendations to improve our manuscript. This study confirmed previous reports indicating a connection between mGluR5 and mitochondrial function, while providing new evidence and novel mitochondrial features regulated by mGluR5, i.e. specific protein levels, mitochondrial morphology and NADP/H level. Moreover, additional hypotheses for follow-up studies can be generated from the proteomic data provided, including studying the possible mechanisms underlying the mitochondrial modulation.

Following the reviewer’s advice, we now discussed previous findings, the novelty of the paper and added the suggested references on page 14 line 536-556, and highlighted text, as: “In previous studies, indications of the association between mGluR5 and mitochondrial function have been found. MGluR5 was shown to modulate ATP levels in different types of cells [74–76], production of ROS [15,72,75,77], mitochondrial mass [75] and mitochondrial biogenesis [78]. Our study revealed that mGluR5, in addition to these, can also regulate specific mitochondrial proteins, morphology and NADP/H level. Our observation of an mGluR5-dependent postsynaptic mitochondrial morphology may partly explain the dysfunctional mitochondrial respiration and morphology recently reported in a fragile X syndrome mouse model [79], in which mGluR5 plays a predominant role [80]. Moreover, the proteomic data provides a new entry into exploration of the specific mitochondrial pathways implicated and the mechanisms by which mitochondrial function responds to mGluR5. Among the multiple downstream effectors of mGluR5 signaling, PKC beta (Prkcb) level was found increased in mGluR5 KO, and MEK1 (Map2k1) level decreased in CTEP-treated mice. These observations suggest the involvement of Gq-IP3-PKC rather than MEK-ERK signaling pathway in the mGluR5-mediated mitochondrial regulation, which might be considered for further investigation. Accordingly, mitochondrial ROS have been found activated by mGluR5 through IP3 [15], and mitochondrial respiration in synaptic fractions is known to be altered via PKC and IP3 in ischemia [81–83]. Taken together, we found new evidence for the connection between mGluR5 and mitochondrial function, and provided data to generate hypotheses for future investigation regarding the underlying mechanistic link between mGluR5 and mitochondria, as well as the functional and therapeutic implication of this relation.”.

Line 350: The subtitle of this section is about “pharmacological” inhibition of mGluR5, however, the results of mGluR5 KO model were also described.

We now corrected this section subtitle as “Reduced mGluR5 activity alters specific mitochondrial pathways“.

Reviewer 3 Report

Gonzales-Lozano et al. showed nicely that blocking of mGluR5 activity, either by genetic or pharmacological means, regulates neuronal mitochondrial function. In a well-designed and methodological powerful study, namely in terms of proteomics, authors provide strong evidence supporting a role for the metabotropic glutamate receptor type 5 in regulating the mitochondrial function in neurons. Overall. I strong support the publication of this study at Cells, and I only have minor revision suggestions.

  1. Authors used a synaptosomal preparation to look at changes at protein expression by MS, however the functional assays were performed in total hippocampal extracts. Therefore, given that they did not investigate mitochondrial function at the synaptic level, I think authors should change the title accordingly.

  1. Can authors please define SWATCH? Line 229.

  1. Fig. 1: it will help the understanding of the figure if the protein databases included in Fig. 1B (e.g. WP) are defined in the respective legend of the figure. Also, the method to check the protein loading should be indicated in the legend of the figure. In the quantification of the western blot data, did authors normalize the levels of their proteins of interest by this loading control?

  1. I think it would be more rigorous if authors refer to western blot as a semi-quantitative method, line 267.

  1. Fig 2: I think it would work better if mitochondria are indicated in the representative image (Fig. 2A), either with some color or with arrowheads, for instance. Also, in the legend, authors indicate as statistical test: t-test (e.g. Fig. 2B) or nested (e.g. Fig. 2C). In the methods section, it seems that authors used a nested test. Which statistical test did authors use to analyze the different panels of Fig. 2 (B-F)? Did they use the same statistical test or not?

  1. In the context of the Fig. 2, but also for the other datasets, did authors test for normal distribution? Which statistical test was used for this analysis? This information should be added to the summary of the statistical analysis in the methods section.

  1. It is not clear, at least for me, what do authors mean with “52 proteins were significantly regulated in at least one group”, lines: 320-231. Can authors be more explicit here please?

  1. Authors refer to panel Fig. 3C to describe their gene ontology analysis of Fig. 3, but this panel does not show any gene ontology analysis (volcano plot of 24 hr treatment). Can authors please indicate the right panel here?

  1. Finally, it would be nice if authors discuss the possible mechanism(s) that mGluR5 uses to modulate the expression of mitochondrial proteins at the synapse, and the neuronal mitochondrial function. The current version of the discussion lacks their view on the mGluR5-mediated mechanism(s) on mitochondrial protein expression and function in neurons and synapses.

Author Response

Gonzales-Lozano et al. showed nicely that blocking of mGluR5 activity, either by genetic or pharmacological means, regulates neuronal mitochondrial function. In a well-designed and methodological powerful study, namely in terms of proteomics, authors provide strong evidence supporting a role for the metabotropic glutamate receptor type 5 in regulating the mitochondrial function in neurons. Overall. I strong support the publication of this study at Cells, and I only have minor revision suggestions.

 We appreciate the kind words and suggestions of the reviewer to improve our manuscript.

  1. Authors used a synaptosomal preparation to look at changes at protein expression by MS, however the functional assays were performed in total hippocampal extracts. Therefore, given that they did not investigate mitochondrial function at the synaptic level, I think authors should change the title accordingly.

We understand the point of the reviewer and we have modified the title accordingly as “Reduced mGluR5 activity modulates mitochondrial function”.

  1. Can authors please define SWATCH? Line 229.

We now defined the abbreviation SWATH in line 125 and line 235 as “Sequential Window Acquisition of all THeoretical mass spectra”.

  1. 1: it will help the understanding of the figure if the protein databases included in Fig. 1B (e.g. WP) are defined in the respective legend of the figure. Also, the method to check the protein loading should be indicated in the legend of the figure. In the quantification of the western blot data, did authors normalize the levels of their proteins of interest by this loading control?

We thank the reviewer for the comment. Differences in protein loading were corrected using the quantification of the total protein amount. Then, protein levels quantified by immunoblotting were normalized to the control (wild-type) group. We now included this information on page 3 (line 147) and Fig.1 legend. Additionally, we indicated in the figure legend the method used for the visualization of the protein loading (2,2,2-Trichloroethanol) and the databases shown in panel B.

  1. I think it would be more rigorous if authors refer to western blot as a semi-quantitative method, line 267.

We simplified our statement on page 6 line 276 as: “As an independent validation of the results obtained by mass spectrometry, we performed immunoblotting analysis of several proteins and with different directions of regulation (Fig. 1B).”.

  1. Fig 2: I think it would work better if mitochondria are indicated in the representative image (Fig. 2A), either with some color or with arrowheads, for instance. Also, in the legend, authors indicate as statistical test: t-test (e.g. Fig. 2B) or nested (e.g. Fig. 2C). In the methods section, it seems that authors used a nested test. Which statistical test did authors use to analyze the different panels of Fig. 2 (B-F)? Did they use the same statistical test or not?

Figure 2A has been updated to highlight the mitochondria in the representative images. Regarding the statistical analysis, in Fig. 2C-F each mitochondrion corresponds to an individual observation and multiple mitochondria were analyzed for each animal; therefore, nested analysis was applied. In Fig. 2B, each animal corresponds to an individual observation and, thus, Student’s t test was used. For clarity, we now indicated both tests in the methods section lines 221-223, and we specified in the figure legend the test and p value for each panel.

  1. In the context of the Fig. 2, but also for the other datasets, did authors test for normal distribution? Which statistical test was used for this analysis? This information should be added to the summary of the statistical analysis in the methods section.

Shapiro-Wilk test and visual inspection were used to test for normality, as now stated on page 5 line 227.

  1. It is not clear, at least for me, what do authors mean with “52 proteins were significantly regulated in at least one group”, lines: 320-231. Can authors be more explicit here please?

We adjusted this sentence to clarify the message as: “In total, 52 proteins were significantly regulated upon CTEP treatments (eBayes p ≤ 0.01, Fig. 3B-C); 45 were found regulated in the CTEP2h group (Fig. 3B), while only 5 proteins were regulated in CTEP1d (Fig. 3C) and 3 in CTEP2d (Fig. 3D).”.

  1. Authors refer to panel Fig. 3C to describe their gene ontology analysis of Fig. 3, but this panel does not show any gene ontology analysis (volcano plot of 24 hr treatment). Can authors please indicate the right panel here?

We thank the reviewer for pointing this out. The main text has been modified to refer in line 349 to the correct panel (Fig. 3E).

  1. Finally, it would be nice if authors discuss the possible mechanism(s) that mGluR5 uses to modulate the expression of mitochondrial proteins at the synapse, and the neuronal mitochondrial function. The current version of the discussion lacks their view on the mGluR5-mediated mechanism(s) on mitochondrial protein expression and function in neurons and synapses.

Following the reviewer’s advice, we now expanded the discussion on page 14 (lines 544-556) to include an hypothesis of the possible mechanisms underlying the mGluR5-mediated mitochondrial modulation: “Moreover, the proteomic data provides a new entry into exploration of the specific mitochondrial pathways implicated and the mechanisms by which mitochondrial function responds to mGluR5. Among the multiple downstream effectors of mGluR5 signaling, PKC beta (Prkcb) level was found increased in mGluR5 KO, and MEK1 (Map2k1) level decreased in CTEP-treated mice. These observations suggest the involvement of Gq-IP3-PKC rather than MEK-ERK signaling pathway in the mGluR5-mediated mitochondrial regulation, which might be considered for further investigation. Accordingly, mitochondrial ROS have been found activated by mGluR5 through IP3 [15], and mitochondrial respiration in synaptic fractions is known to be altered via PKC and IP3 in ischemia [81–83]. Taken together, we found new evidence for the connection between mGluR5 and mitochondrial function, and provided data to generate hypotheses for future investigation regarding the underlying mechanistic link between mGluR5 and mitochondria, as well as the functional and therapeutic implication of this relation.”.
